# Local orthorhombic lattice distortions in the paramagnetic tetragonal phase of superconducting $NaFe_{1−x}Ni_xAs$

Weiyi Wang[1], Yu Song[1], Chongde Cao[2], Kuo-Feng Tseng[3,4], Thomas Keller [3,4], Yu Li[1],
L.W. Harriger[5], Wei Tian[6], Songxue Chi[6], Rong Yu[7], Andriy H. Nevidomskyy[1] & Pengcheng Dai [1]

Understanding the interplay between nematicity, magnetism and superconductivity is pivotal for elucidating the physics of iron-based superconductors. Here we use neutron scattering to probe magnetic and nematic orders throughout the phase diagram of $NaFe_{1−x}Ni_xAs$, finding that while both static antiferromagnetic and nematic orders compete with superconductivity, the onset temperatures for these two orders remain well separated approaching the putative quantum critical points. We uncover local orthorhombic distortions that persist well above the tetragonal-to-orthorhombic structural transition temperature $T_s$ in underdoped samples and extend well into the overdoped regime that exhibits neither magnetic nor structural phase transitions. These unexpected local orthorhombic distortions display Curie–Weiss temperature dependence and become suppressed below the superconducting transition temperature $T_c$, suggesting that they result from the large nematic susceptibility near optimal superconductivity. Our results account for observations of rotational symmetry breaking above $T_s$, and attest to the presence of significant nematic fluctuations near optimal superconductivity.

[1] Department of Physics and Astronomy, Rice University, Houston, TX 77005, USA. [2] Department of Applied Physics, Northwestern Polytechnical University, Xian 710072, China. [3] Max-Planck-Institut für Festkörperforschung, Heisenbergstrasse 1, D-70569 Stuttgart, Germany. [4] Max Planck Society Outstation at the Forschungsneutronenquelle Heinz Maier-Leibnitz (MLZ), D-85747 Garching, Germany. [5] NIST Center for Neutron Research, National Institute of Standards and Technology, Gaithersburg, MD 20899, USA. [6] Neutron Scattering Division, Oak Ridge National Laboratory, Oak Ridge, TN 37831, USA. [7] Department of Physics and Beijing Key Laboratory of Opto-electronic Functional Materials and Micro-nano Devices, Renmin University of China, Beijing 100872, China. These authors contributed equally: Weiyi Wang, Yu Song. Correspondence and requests for materials should be addressed to C.C. (email: caocd@nwpu.edu.cn) or to P.D. (email: pdai@rice.edu)

Iron pnictide superconductors are a large class of materials hosting unconventional superconductivity that emerges from antiferromagnetically ordered parent compounds [Fig. 1a]. Unique to iron pnictides is the tetragonal-to-orthorhombic structural transition at $T_s$, where the underlying lattice changes from exhibiting fourfold ($C_4$) above $T_s$ to twofold ($C_2$) rotational symmetry below $T_s$, which occurs either simultaneously with or above the antiferromagnetic (AF) phase transition temperature $T_N$ [Fig. 1b][1,2]. The large electronic anisotropy present in the paramagnetic orthorhombic phase has been ascribed to an electronic nematic state[3–5] that couples with the shear strain of the lattice, the orthorhombicity $\delta$ [$=(a-b)/(a+b)$, where $a$ and $b$ are in-plane orthorhombic lattice parameters], therefore acts as a proxy for the nematic order parameter. In the paramagnetic tetragonal state, the nematic susceptibility can be measured via determining the resistivity anisotropy induced by anisotropic in-plane strain[6] or by measuring the elastic shear modulus[7,8]. By fitting temperature dependence of nematic susceptibility with a Curie–Weiss form, a nematic quantum critical point (QCP) with Weiss temperature $T^* \to 0$ has been identified near optimal superconductivity for different iron-based superconductors[6,8]. Theoretically, the proliferation of nematic fluctuations near the nematic QCP can act to enhance Cooper pairing[9–12].

Although $C_4 \to C_2$ symmetry breaking is typically associated with the structural transition at $T_s$, there are numerous reports of its observation well above $T_s$ and in overdoped compounds[13–19]. These observations are either reflective of an intrinsic rotational symmetry-broken phase above $T_s$, which can occur in bulk[13–15] or on the surface of the sample[16], or simply a result from a large nematic susceptibility[17–20]. In the first case, there is a small, but nonzero nematic order parameter throughout the material above $T_s$, although no additional symmetry breaking occurs below $T_s$, despite the sharp increase of the nematic order parameter. For the latter scenario, only local orthorhombic distortions can be present and the system remains tetragonal on average. One way to differentiate the two scenarios is to directly and quantitatively probe the distribution of the interplanar atomic spacings ($d$-spacings) and its temperature dependence.

Ideally, when the system becomes orthorhombic, two different in-plane $d$-spacings, corresponding to different in-plane lattice parameters, can be resolved; on the other hand, when there are only local orthorhombic distortions, the $d$-spacing distribution only broadens, while the average structure remains tetragonal [Fig. 1c]. However, experimentally, it can be very difficult to distinguish the two scenarios when $\delta$ is too small for a splitting to be resolved, then, a broadening is also seen even when the system goes through a tetragonal-to-orthorhombic phase transition. In such cases, it is more instructive to examine the temperature dependence of the experimentally obtained broadening, characterized either by $\delta$ or by the width of the $d$-spacing distribution, $\Delta d/d$ [Fig. 1c]. For a phase transition, the broadening should exhibit a clear order parameter-like onset; for local orthorhombic distortions in an average tetragonal structure, the broadening instead tracks the nematic susceptibility, which exhibits a Curie–Weiss temperature dependence[4] [Fig. 1c]. An additional complication is that the AF order typically becomes spin-glass-like and sometimes incommensurate near the nematic QCP[21–25], and given the strong magnetoelastic coupling in iron pnictides[5,8], it is unclear how such changes in AF order affect the nematic order.

In this work, we use high-resolution neutron diffraction and neutron Larmor diffraction to map out the phase diagram of NaFe$_{1-x}$Ni$_x$As[26], focusing on the interplay between magnetic order, nematic order, and superconductivity near optimal superconductivity. Unlike most other iron pnictide systems, we find $T_N$ in NaFe$_{1-x}$Ni$_x$As to be continuously suppressed toward $T_N \approx T_c$ near optimal doping, while the order remains long-range and commensurate. This allows us to demonstrate that $T_s$ and $T_N$ in NaFe$_{1-x}$Ni$_x$As remain well separated near optimal superconductivity, indicating distinct QCPs associated with nematic and AF orders, similar to the quantum criticality in electron-doped Ba$_2$Fe$_{2-x}$Ni$_x$As$_2$[27]. Utilizing the high resolution provided by neutron Larmor diffraction[28,29], we probed the nematic order parameter in underdoped NaFe$_{1-x}$Ni$_x$As below $T_s$ and surprisingly, uncovered local orthorhombic distortions well above $T_s$ and in overdoped samples without a structural phase transition. Although the average structure is tetragonal in these regimes, broadening of the $d$-spacing distribution is unambiguously observed. Such local orthorhombic distortions were hinted in previous high-resolution neutron powder diffraction measurements on electron-overdoped NaFe$_{0.975}$Co$_{0.025}$As, where a small broadening of Bragg peaks at low temperature was observed[26]. Regardless of whether orthorhombic distortions are long-range due to a structural phase transition or local in nature, resulting from large nematic susceptibility, we find that they become suppressed inside the superconducting state, similar to AF order. Our results, therefore, elucidate the interplay between AF order, nematicity, and superconductivity in NaFe$_{1-x}$Ni$_x$As; at the same time, our observation of local orthorhombic distortions with a Curie–Weiss temperature dependence across the phase diagram accounts for rotational symmetry breaking seen in nominally tetragonal iron pnictides. In addition, our measurements demonstrate that neutron Larmor diffraction can be used to determine the nematic susceptibility of free-standing iron pnictides without the need to apply external stress or strain. These results should stimulate future high-resolution neutron/X-ray diffraction work to study orthorhombic lattice distortion and its temperature dependence in the nominally tetragonal phase of iron-based superconductors.

## Results

**Overall phase diagram of NaFe$_{1-x}$Ni$_x$As.** Our results are reported using the orthorhombic structural unit cell with lattice parameters $a \approx b \approx 5.56$ Å and $c \approx 7.05$ Å for NaFeAs[30,31]. The momentum transfer $\mathbf{Q} = H\mathbf{a}^* + K\mathbf{b}^* + L\mathbf{c}^*$ is denoted as $\mathbf{Q} = (H, K, L)$ in reciprocal lattice units (r.l.u.), with $\mathbf{a}^* = \hat{\mathbf{a}}2\pi/a$, $\mathbf{b}^* = \hat{\mathbf{b}}2\pi/b$, and $\mathbf{c}^* = \hat{\mathbf{c}}2\pi/c$. In this notation, magnetic Bragg peaks are at $\mathbf{Q} = (1, 0, L)$, with $L = 0.5, 1.5, 2.5, \ldots$ Samples were mostly aligned in the $[1, 0, 0] \times [0, 0, 1]$ scattering plane, which allows scans of magnetic peaks along $H$ and $L$; the $x = 0.012$ sample was also studied in the $[1, 0, 1.5] \times [0, 1, 0]$ plane. We have carried out neutron diffraction, neutron Larmor diffraction, and inelastic neutron scattering experiments on NaFe$_{1-x}$Ni$_x$As (see Methods section for experimental details).

Figure 1d shows the overall phase diagram determined from our experiments, with $T_s$, $T_N$, and $T_c$ marked. Although for optimal-doped and over-doped regimes, the samples on average exhibit a tetragonal structure at all temperatures, there are local orthorhombic distortions resulting in broadening of $d$-spacing distribution that can be characterized by $\delta$ or $\Delta d/d$. The orthorhombic distortion $\delta$ is plotted in a pseudo-color scheme as a function of temperature and doping near optimal-doping in Fig. 1d. Figure 1e shows the Ni-doping dependence of the ordered magnetic moment and $\delta$ at $T = 5$ K, and $T = T_c$ for superconducting samples. With increasing Ni-doping $x$, the AF ordered moment and $T_N$ decrease monotonically, and no magnetic order is detected in the $x = 0.015$ sample [Fig. 1e]. While the magnetic order parameter for the $x = 0.004$ sample resembles that of NaFeAs [Fig. 2e, f], the magnetic order becomes strongly suppressed upon entering the superconducting state for $x = 0.010$ [Fig. 2g], similar to other iron pnictides[32,33].

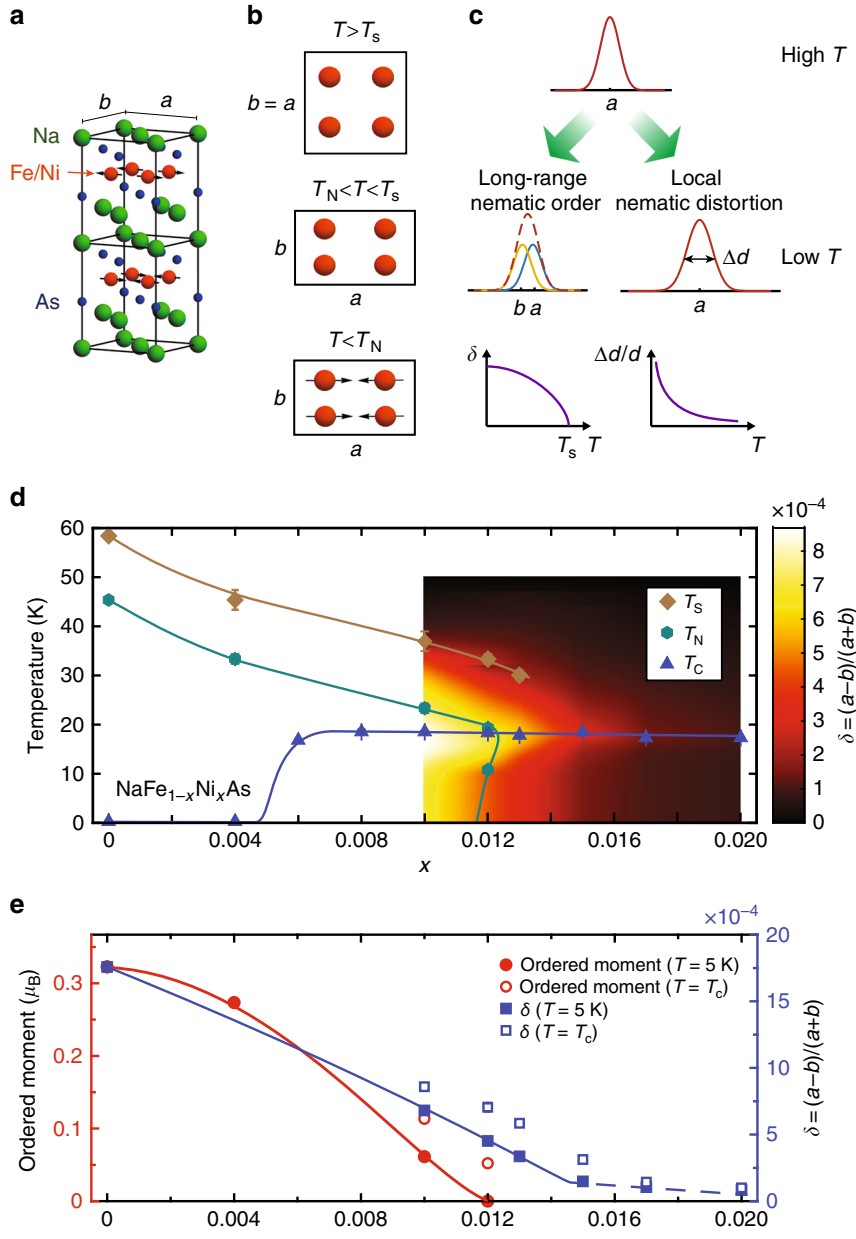

**Fig. 1** The phase diagram of NaFe$_{1-x}$Ni$_x$As determined from neutron scattering measurements. **a** Crystal structure and magnetic order of NaFeAs. **b** Schematic evolution of NaFe$_{1-x}$Ni$_x$As in tetragonal, paramagnetic orthorhombic, and AF orthorhombic states. **c** Schematic of how $d$-spacing distribution changes from a tetragonal state at high temperatures to an orthorhombic state through phase transition [characterized by $\delta = (a - b)/(a + b)$] or a state with local orthorhombic distortions (characterized by broadening of the $d$-spacing distribution $\Delta d/d$) that on an average remains tetragonal. For the orthorhombic state, when the splitting $\delta$ is too small to be resolved experimentally, a broadening is also observed (red dashed line). In such cases, the two situations can nonetheless be differentiated by examining the temperature dependence of $\delta$ or $\Delta d/d$. **d** The phase diagram of NaFe$_{1-x}$Ni$_x$As. $T_s$, $T_N$, and $T_c$ are the transition temperatures for the tetragonal-to-orthorhombic structural phase transition, the AF phase transition, and the superconducting transition. The point for $x = 0$ is obtained from ref.[25]. **e** The Ni-doping dependence of the ordered magnetic moment and orthorhombic distortion $\delta$ at $T = 5$ K and $T = T_c$ for superconducting samples. The error bars in (**d**) are estimated errors from fits-to-order parameters and transition temperatures

**Reentry into the paramagnetic state in NaFe$_{1-x}$Ni$_x$As with $x = 0.012$.** For the $x = 0.012$ sample, magnetic order begins at $T_N \approx$ 19 K and becomes strongly suppressed upon entering the superconducting state below $T_c$ and reenters into the paramagnetic state without any long-range order below $T_r \approx 10$ K [Fig. 2h]. Given the sharp superconducting transition at $T_c$ (Supplementary Fig. 1 and Methods section), $T_r$ is well inside the superconducting state. This is similar to the behavior of nearly

optimal-doped Ba(Fe$_{0.941}$Co$_{0.059}$)$_2$As$_2$[34], although AF order in Ba(Fe$_{0.941}$Co$_{0.059}$)$_2$As$_2$ is short-range and incommensurate[21]. To confirm that the magnetic order in our $x = 0.012$ sample is long-range and commensurate, we carried out scans along [H, 0, 1.5], [1, K, 1.5] and [1, 0, L] directions in [1, 0, 1.5] × [0, 1, 0] and [1, 0, 0] × [0, 0, 1] scattering planes [Fig. 2a], with results summarized in Fig. 2b–d. As can be seen, magnetic order remains long-range along all three high-symmetry directions (with spin–spin

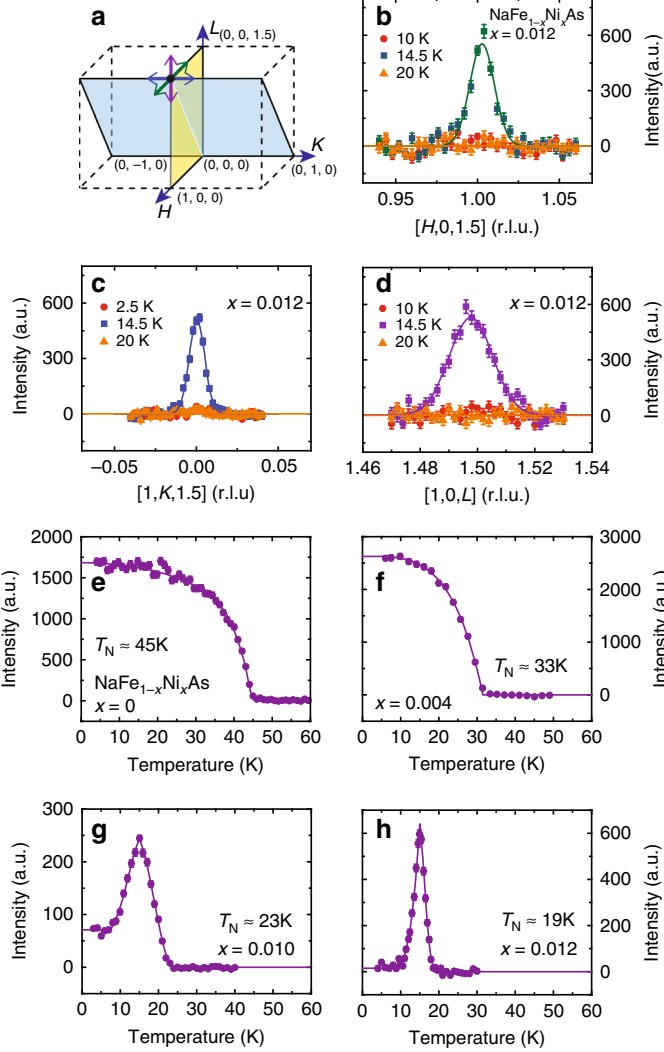

**Fig. 2** Neutron scattering geometry and doping dependence of the magnetic order parameter for NaFe$_{1-x}$Ni$_x$As. **a** Schematic of [1, 0, 0] × [0, 0, 1] and [1, 0, 1.5] × [0, 1, 0] scattering planes that allow scans of the magnetic Bragg peak **Q** = (1, 0, 1.5) along (**b**) [H, 0, 1.5], (**c**) [1, K, 1.5], and (**d**) [1, 0, L] directions. Magnetic order parameters measured at **Q** = (1, 0, 1.5) for NaFe$_{1-x}$Ni$_x$As with (**e**) x = 0, (**f**) x = 0.004, (**g**) x = 0.010, and (**h**) x = 0.012. No magnetic order is observed for x = 0.015. Data in (**e**) are from ref.[25]. All vertical error bars in the figure represent statistical errors of 1 s.d

correlation lengths exceeding 100 Å) for the x = 0.012 sample near optimal superconductivity before disappearing near x = 0.015. These wave–vector scans also confirm the complete disappearance of long-range magnetic order below $T_r$. For comparison, we note that magnetism in electron-doped Ba(Fe$_{1-x}$Co$_x$)$_2$As$_2$ (~6%)[21], BaFe$_{2-x}$Ni$_x$As$_2$ (~5%)[22], and NaFe$_{1-x}$Co$_x$As (~2.3%)[25] exhibits cluster spin glass and incommensurate magnetic order near optimal superconductivity likely related to impurity effects[23,35]. The absence of such behavior in NaFe$_{1-x}$Ni$_x$As is likely a result of significantly lower dopant concentration in NaFe$_{1-x}$Ni$_x$As (~1.3%) near optimal doping. Our inelastic neutron scattering measurements on the x = 0.012 sample confirm that the presence of a neutron spin resonance, which can act as a proxy for the superconducting order parameter, is unaffected when cooled below $T_r$ (Supplementary Fig. 2 and Methods section).

**Nematic order and local orthorhombic distortions in NaFe$_{1-x}$Ni$_x$As**. Having established the evolution of AF order and its interplay with superconductivity in NaFe$_{1-x}$Ni$_x$As, we examined the Ni-doping evolution of the nematic order in NaFe$_{1-x}$Ni$_x$As. To precisely determine the evolution of orthorhombic distortion, we used high-resolution neutron diffraction and neutron Larmor diffraction to investigate the temperature evolution of the orthorhombic lattice distortion (Supplementary Figs. 3 and 4 and Methods section). For NaFe$_{1-x}$Ni$_x$As with x ≤ 0.013, we can see clear orthorhombic lattice distortion below $T_s$, also confirmed by the anomalies in temperature dependence of electrical resistivity measurements (Supplementary Fig. 5 and Methods section). Figure 3a–c shows temperature and Ni-doping dependence of the orthorhombic distortion δ. For NaFe$_{1-x}$Ni$_x$As with x ≤ 0.013 at temperatures above $T_s$, and for x ≥ 0.015 at all temperatures, the system is on an average tetragonal and should in principle have δ = 0. Surprisingly, we see clear temperature-dependent δ. Moreover, while δ below $T_s$ behaves as expected for an order parameter associated with phase transition, δ in temperature regimes with an average tetragonal structure exhibits a Curie–Weiss temperature dependence, suggesting that it arises from local orthorhombic distortions. In all cases, we find that δ decreases dramatically below $T_c$, indicating that orthorhombic distortion, whether long-range or local, competes with superconductivity. The competition between superconductivity and long-range nematic order is similar to Ba(Fe$_{1-x}$Co$_x$)$_2$As$_2$[36] and can be captured by a phenomenological Landau theory, based on an effective action in terms of the corresponding order parameters (see Methods section):

$$F[\Delta, \delta] = \frac{C}{2}\delta^2 + \frac{D}{4}\delta^4 - \frac{\alpha}{2}|\Delta|^2 + \frac{\beta}{4}|\Delta|^4 + \gamma|\Delta|^2\delta^2, \quad (1)$$

where, the last term describes the competition between nematicity and superconductivity. As a result, the nematic order parameter is noticeably suppressed inside the superconducting phase, compared with its value ($\delta_0$) in the normal phase, so that (see Methods section for the derivation)

$$\delta^2 \simeq \delta_0^2 - \left(\frac{2\gamma}{D}\right)|\Delta|^2, \quad (2)$$

whereas the superconducting order parameter itself remains essentially unchanged due to tiny values of $\delta_0$ (see Eq. (8) in Methods section). In the tetragonal phase (δ = 0), the competition between local orthorhombic distortions and superconductivity is reflective of the suppression of nematic susceptibility below $T_c$[37].

We emphasize that the local orthorhombic distortions we uncovered in the tetragonal phase of NaFe$_{1-x}$Ni$_x$As are distinct from the phase separation into superconducting tetragonal and AF orthorhombic regions found in Ca(Fe$_{1-x}$Co$_x$)$_2$As$_2$ under biaxial strain[38,39]. In the latter compound, the quantum phase transition between the superconducting tetragonal and AF orthorhombic phases is of first order, and the resulting phase separation into these two phases with different in-plane lattice parameters allows the material to respond to biaxial strain in a continuous fashion; this would occur even if there were no quenched disorder. In NaFe$_{1-x}$Ni$_x$As, the quantum phase transition is of second order and, therefore an analogous phase separation does not occur. Instead, the local orthorhombic distortions we observe in NaFe$_{1-x}$Ni$_x$As likely result from large nematic susceptibility near optimal superconductivity pinned by quenched disorder.

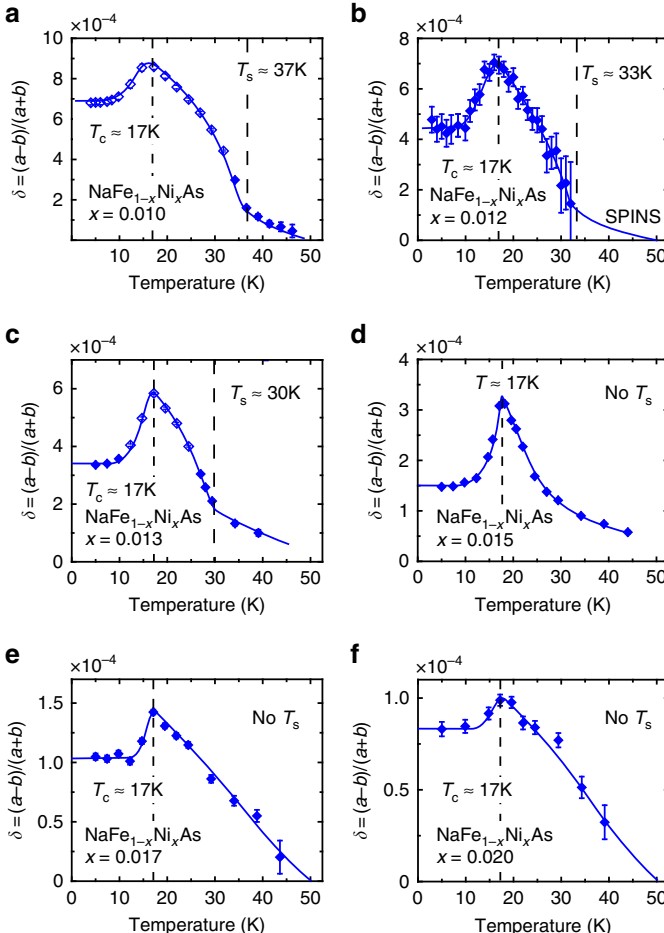

**Fig. 3** Neutron diffraction and neutron Larmor diffraction studies of Ni-doping dependence of the orthorhombic distortion in $NaFe_{1-x}Ni_xAs$. Temperature dependence of the orthorhombic distortion $\delta$ for $NaFe_{1-x}Ni_xAs$ with (**a**) $x = 0.01$, (**b**) $x = 0.012$, (**c**) $x = 0.013$, (**d**) $x = 0.015$, (**e**) $x = 0.017$, and (**f**) $x = 0.02$. Data in (**b**) are obtained from high-resolution neutron diffraction, whereas all the other panels are obtained from neutron Larmor diffraction measurements. Solid lines are guides to the eye. $\delta$ is obtained by assuming that it is 0 at $T = 50$ K, and broadening at lower temperatures are fit with two split peaks, with widths of the single peak at $T = 50$ K. Open symbols correspond to measurements where a splitting is definitively observed, and solid symbols represent measurements that only resolve a broadening due to experimental limitations (Methods section and Supplementary Fig. 4). All vertical error bars in the figure represent least-square fits to the raw data with errors of 1 s.d

Given that the orthorhombic distortion with Curie–Weiss temperature dependence arises from local orthorhombic distortions, an alternative way to characterize such distortion is broadening of $d$-spacing distribution width, $\Delta d/d$ (see Methods section). In Fig. 4a–d, we show $\Delta d/d$ in $NaFe_{1-x}Ni_xAs$, obtained from our neutron Larmor diffraction measurements. Given that the local orthorhombic distortions arise from quenched disorder coupled with large nematic susceptibility near a nematic QCP, it should track the temperature dependence of nematic susceptibility, since the quenched disorder should depend weakly on temperature. Therefore, we have fitted $\Delta d/d$ in Fig. 4a–d with the Curie–Weiss form $\Delta d/d \propto 1/(T - T^*)$ and extracted the Weiss temperature $T^*$ as a function of doping, as shown in Fig. 4e. Our $\Delta d/d$ results are well described by the Curie–Weiss form, with $T^*$ changing from positive in underdoped to negative in overdoped regime [Fig. 4e], suggesting a nematic QCP near optimal

superconductivity. These results are reminiscent of temperature and doping dependence of nematic susceptibility from elastore-sistance[6] and shear modulus measurements[8], suggesting that temperature dependence of $\Delta d/d$ is a direct measure of the nematic susceptibility without the need to apply external stress.

## Discussion

In $NaFe_{1-x}Ni_xAs$, the orthorhombic distortion and the structural phase transition temperature are $\delta \approx 1.7 \times 10^{-3}$ and $T_s \approx 58$ K for $x = 0$[25,31]; for $x = 0.012$, they become $\delta \approx 7 \times 10^{-4}$ and $T_s \approx 33$ K. We find no evidence of structural phase transitions for samples with $x \geq 0.015$, suggesting the presence of a putative nematic QCP at $x = x_c$, where $x_c \gtrsim 0.015$. These results are consistent with recent Muon spin rotation and relaxation study of the magnetic phase diagram of $NaFe_{1-x}Ni_xAs$[40]. The doping-dependence of $T_s$ and $\delta$ are also consistent with the Ni-doping dependence of $T^*$ determined from Curie–Weiss fits to temperature dependence of $\Delta d/d$, which changes from positive to negative near $x \approx 0.015$ [Fig. 4e]. Since our neutron Larmor diffraction measurements were carried out using polarized neutron beam produced by an Heusler monochromator, which has an energy resolution of about $\Delta E \approx 1.0$ meV[28,29], the local orthorhombic distortions captured in our measurements are either static or fluctuating slower than a time scale of $\tau \sim \hbar/2\Delta E \sim 0.3$ ps, where $\hbar$ is the reduced Planck's constant[41,42]. One possible origin of such slow fluctuations may be the in-plane transverse acoustic phonons that exhibit significant softening in the paramagnetic tetragonal phase when approaching a nematic instability[43]. Future neutron scattering experiments with energy resolutions much better than $\Delta E \approx 1$ meV are desirable to separate the static and slowly fluctuating contributions. Our results also indicate that the nematic QCP would occur at a $x$ value that is distinctively larger than that of the magnetic QCP in the absence of superconductivity. In the phase diagram of iron pnictides with decoupled $T_s$ and $T_N$, due to the competition between superconductivity with both nematic and magnetic orders, magnetic order forms a hump peaked at $T_c$ near optimal doping [Fig. 1d], and the structural phase transition disappears in a similar fashion at a larger $x$.

Theoretically, a determinantal quantum Monte Carlo study of a two-dimensional sign-problem-free lattice model reveals an Ising nematic QCP in a metal at finite fermion density[44]. In the nematic phase, the discrete lattice rotational symmetry is spontaneously broken from fourfold to twofold, and there are also nematic quantum critical fluctuations above the nematic ordering temperature. Within the numerical accuracy of the determinantal quantum Monte Carlo study, the uniform nematic susceptibility above the nematic ordering temperature has Curie–Weiss temperature dependence, signaling an asymptotic quantum critical scaling regime consistent with our observation[44]. Alternatively, the observed Curie–Weiss temperature-dependent behavior of nematic susceptibility can be understood from spin-driven nematic order theory, where magnetic fluctuations associated with static AF order induce formation of the nematic state[45]. In this picture, the effect of lattice strain coupled to the nematic order parameter produces a mean-field Curie–Weiss-like behavior, arising from the nemato-elastic coupling which has direction-dependent terms in the propagator for nematic fluctuations. The Curie–Weiss temperature-dependent nematic susceptibility should occur in the entire phase diagram, where there is a significant softening of the elastic modulus[45]. This means that Curie–Weiss temperature dependence of local orthorhombic distortions that we observe is a signature of nemato-elastic coupling, which does not suppress the magnetic fluctuations that cause the nematic order, but transforms the Ising nematic transition into a mean-field transition[45].

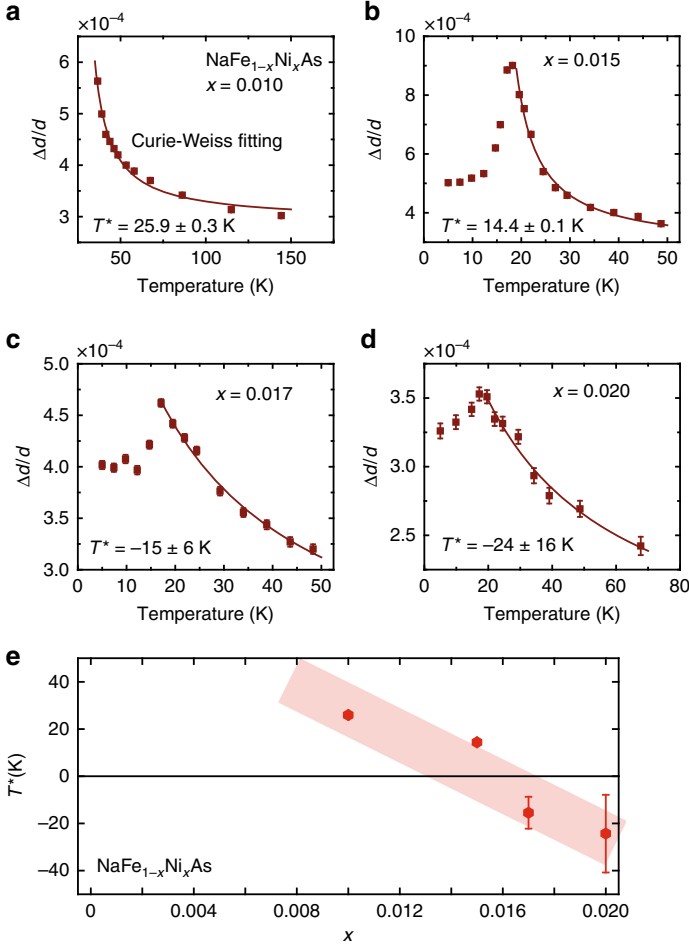

**Fig. 4** Curie–Weiss fit to temperature dependence of $\Delta d/d$ as a function of $x$ for NaFe$_{1-x}$Ni$_x$As. Temperature dependence of $\Delta d/d$ and Curie–Weiss fit to the data for (**a**) $x = 0.01$, (**b**) $x = 0.015$, (**c**) $x = 0.017$, and (**d**) $x = 0.017$. Clear reduction below $T_c$ is seen for all tetragonal samples. **e** Ni-doping dependence of Weiss temperature $T^*$, showing a change of sign around $x = 0.015$, suggesting the presence of a nematic QCP. Data points in (**a–d**) with $T < 50$ K are obtained from the same neutron Larmor diffraction data used to extract $\delta$ in Fig. 3 (see Methods Section). All vertical error bars in the figure represent least-square fits to the raw data with errors of 1 s.d

Our discovery of local orthorhombic distortions exhibiting Curie–Weiss temperature dependence across the phase diagram of NaFe$_{1-x}$Ni$_x$As results from the proliferation of nematic fluctuations and large nematic susceptibility near the nematic QCP. Quenched disorder that are always present in such doped materials act to pin the otherwise fluctuating local nematic domains, resulting in static (or quasi-static) local orthorhombic distortions that can lead to observations of rotational symmetry breaking seen with multiple probes[13–19]. We have definitively observed local nematic distortions in NaFe$_{1-x}$Ni$_x$As that are static or quasi-static, in contrast to local distortions seen in Sr$_{1-x}$Na$_x$Fe$_2$As$_2$, using pair distribution function analysis that contain significantly more dynamic contributions[46] and which would not cause rotational symmetry breaking seen by static probes. Our observation of local nematic distortions highlights the presence of nematic fluctuations near the nematic QCP, which can play an important role in enhancing superconductivity of iron pnictides[9–12], while the intense Ising nematic spin correlations near the nematic QCP may be the dominant pairing interaction[47–49].

## Methods

**Elastic neutron scattering experimental details.** Elastic neutron experiments were carried out on the Spin Polarized Inelastic Neutron Spectrometer (SPINS) at the NIST Center for Neutron Research, United States and the HB-1A triple-axis spectrometer at the High-Flux-Isotope Reactor (HFIR), Oak Ridge National

Laboratory (ORNL), United States. We used pyrolytic graphite [PG(002)] monochromators and analyzers in these measurements. At HB-1A, the monochromator is vertically focused with fixed-incident neutron energy $E_i = 14.6$ meV and the analyzer is flat. At SPINS, the monochromator is vertically focused and the analyzer is flat with fixed-scattered neutron energy $E_f = 5$ meV. A PG filter was used at HB-1A and a Be filter was used at SPINS to avoid contamination from higher-order neutrons. Collimations of 40′–40′-sample-40′–80′ and guide-40′-sample-40′-open were used on HB-1A and SPINS, respectively.

To measure the structural distortion in NaFe$_{1-x}$Ni$_x$As ($x = 0.012$) at SPINS, we changed the collimation to guide-20′-sample-20′-open to improve the resolution and removed the Be filter. Our measurement was carried out nominally around a weak nuclear Bragg peak $\mathbf{Q} = (2, 0, 0)$, but the measured intensity at this position mostly come from higher-order neutrons [$\mathbf{Q} = (4, 0, 0)$ for $\lambda/2$ neutrons and $\mathbf{Q} = (6, 0, 0)$ for $\lambda/3$ neutrons]. While we do not resolve two split peaks in the orthorhombic state, clear broadening can be observed. Typical scans along the [$H$, 0, 0] direction centered at $\mathbf{Q} = (2, 0, 0)$ are shown in Supplementary Fig. 3. $\delta$ in Fig. 3b is obtained by assuming $\delta = 0$ at $T = 50$ K and fitting broadening at lower temperatures as two split peaks with fixed widths of the peak at $T = 50$ K.

**Inelastic neutron scattering experimental details.** Our inelastic neutron scattering experiment was carried out on the HB-3 triple-axis spectrometer at HFIR, ORNL, United States. Vertically focused pyrolytic graphite [PG(002)] monochromator and analyzer with fixed-scattered neutron energy $E_f = 14.7$ meV were used. A PG filter was used to avoid higher-order neutron contaminations. The collimation used was 48′-40′-sample-40′-120′.

Using inelastic neutron scattering, we studied the neutron spin resonance mode[2,50] in NaFe$_{1-x}$Ni$_x$As, with $x = 0.012$. Energy scans at $\mathbf{Q} = (1, 0, 0.5)$ above ($T = 35$ K) and below $T_c$ ($T = 1.5$ and 9 K) are compared in Supplementary Fig. 2a. The scans below $T_c$ after subtracting the $T = 35$ K scan are compared in Supplementary Fig. 2b. A clear resonance mode at $E_r = 7$ meV similar to optimal-

doped NaFe$_{1-x}$Co$_x$As[51] is observed, with almost identical intensities at $T = 1.5$ and 9 K. Constant energy scans along [$H$,0,0.5] at different temperatures are compared in Supplementary Fig. 2c, confirming the results in Supplementary Figs. 2a, b. Temperature dependence of the resonance mode is shown in Supplementary Fig. 2d, over-plotted with temperature dependence of orthorhombicity and AF order parameter. Intensity of the resonance mode increases smoothly below $T_c$ and $T_r$, displaying no response when AF order is completely suppressed below $T_r$. These results demonstrate the coexistence of robust superconductivity and nematic order without AF order in NaFe$_{1-x}$Ni$_x$As ($x = 0.012$) below $T_r$.

**Larmor diffraction experimental details**. Our neutron Larmor diffraction measurements were carried out at the three axes spin-echo spectrometer at Forschungs-Neutronenquelle Heinz Maier-Leibnitz (MLZ), Garching, Germany. Neutrons are polarized by a super-mirror bender, and higher-order neutrons are eliminated using a velocity selector. We used double-focused PG(002) monochromator and horizontal-focused Heusler (Cu$_2$MnAl) analyzer in these measurements. Incident and scattered neutron energies are fixed at $E_i = E_f = 15.67$ meV ($k_i = k_f = 2.750$ Å$^{-1}$).

The detailed principles of neutron Larmor diffraction has been described in detail elsewhere[29,52]. In such experiments, polarization of the scattered neutrons $P$ is measured as a function of the total Larmor precession phase $\phi_{tot}$. By analyzing the measured $P(\phi_{tot})$, information about the sample's $d$-spacing distribution can be obtained.

For an ideal crystal with $d$-spacing described by a $\delta$ function, $P$ is independent of $\phi_{tot}$, with $P(\phi_{tot}) = P_0$. $P_0$ accounts for the non-ideal polarization of neutrons and can be corrected for by Ge crystal calibration measurements. In real materials, due to internal strain and sample inhomogeneity, or in the case of iron pnictides, a twinned orthorhombic phase, the $d$-spacing should instead be described by a distribution $f(\epsilon)$, with $\epsilon = \delta d/d$. $\delta d$ is the deviation from the average $d$-spacing $d$. $P(\phi_{tot})$ is then described by:

$$P(\phi_{tot}) = P_0 \int_{-\infty}^{\infty} f(\epsilon)\cos(\phi_{tot}\epsilon)\mathrm{d}\epsilon. \quad (3)$$

Thus, $P(\phi_{tot})$ can be regarded as the Fourier transform of the lattice $d$-spacing distribution $f(\epsilon)$. By measuring $P(\phi_{tot})$, it is possible to resolve features with a resolution better than $10^{-5}$ in terms of $\epsilon$, limited by the range of accessible $\phi_{tot}$.

The distribution of $d$-spacing $f(\epsilon)$ is commonly described as a Gaussian function with full-width-at-half-maximum (FWHM) $\epsilon_{FWHM}$, also denoted as $\Delta d/d$ in the rest of the paper. Eq. (3) then becomes

$$P(\phi_{tot}) = P_0 \exp\left(-\frac{\epsilon_{FWHM}^2}{16\ln 2}\phi_{tot}^2\right). \quad (4)$$

In iron pnictides with a nonzero nematic order parameter, due to twinning, $f(\epsilon)$ becomes the sum of two Gaussian functions. Assuming that the two Gaussian peaks have identical FWHM $\epsilon_{FWHM}$, Eq. (4) becomes

$$P(\phi_{tot}) = P_0 \exp\left(-\frac{\epsilon_{FWHM}^2}{16\ln 2}\phi_{tot}^2\right) \times \sqrt{r^2 + (1-r)^2 + 2r(1-r)\cos(\phi_{tot}\Delta\epsilon)}, \quad (5)$$

where, $r$ and $(1-r)$ denotes the relative populations of the two lattice $d$-spacings $a$ and $b$, and $\Delta\epsilon = 2(a-b)/(a+b) = 2\delta$[53]. Therefore, the nematic order parameter can be extracted by fitting $P(\phi_{tot})$ using Eq. (5).

When $\delta$ is too small to be directly resolved by Larmor diffraction, $P(\phi_{tot})$ can be well described by either Eq. (4) or (5). In such cases, we either extract $\Delta d/d$ from Eq. (4) (Fig. 4) or extract $\delta$ by assuming at $T = 50$ K, $\delta = 0$ and extract $\epsilon_{FWHM}$, then fit to Eq. (5) by fixing $\epsilon_{FWHM}$ to this value (Figs. 1e, 3). Measurements of $P(\phi_{tot})$ at several different temperatures for NaFe$_{1-x}$Ni$_x$As ($x = 0.013$) are shown in Supplementary Fig. 4, and fit to Eq. (5) as described.

A key feature of Eq. (5) is an oscillation in $P(\phi_{tot})$, which can be seen in raw data in Supplementary Fig. 4d–i (open symbols in Fig. 3c); in these cases, the measurement provides definitive evidence of an orthorhombic state. For other panels in Supplementary Fig. 4, due to limited range of $\phi_{tot}$, $P(\phi_{tot})$ can be equally well described by Eq. (4) (solid symbols in Fig. 3c); for such data, we cannot differentiate between a true splitting and a broadening from measurements done at a single temperature.

**Magnetic susceptibility and electrical resistivity measurements**. To ensure that $T_r$ for NaFe$_{1-x}$Ni$_x$As ($x = 0.012$) is well inside the superconducting state, we show in Supplementary Fig. 1 its magnetic susceptibility as a function of temperature. As can be seen, the sample displays a sharp superconducting transition at $T_c \approx 17$ K, with a width $\Delta T_c \approx 2$ K. $T_r$ is well inside the superconducting state, unaffected by the width of the superconducting transition.

The temperature and doping dependence of the in-plane electrical resistivity $\rho(T)$ were measured using the standard four-probe method, the results are normalized to $\rho(200$ K) and summarized in Supplementary Fig. 5. The superconducting transitions for all measured samples are sharp. The kinks associated with the structural transition at $T_s$ can be clearly identified in

underdoped samples (Supplementary Fig. 5a–d), similar to NaFe$_{1-x}$Cu$_x$As[54]. These kinks are progressively suppressed with increasing Ni concentration and disappear in overdoped samples. $T_s$ determined from electrical resistivity measurements are in good agreement with those obtained from Larmor diffraction.

**Coexistence of superconductivity with lattice nematicity**. Here, we first consider the case without any long-range magnetic order, as is realized in NaFe$_{1-x}$Ni$_x$As for $x > 0.012$. In that case, the effective Landau free energy can be written in terms of only the superconducting order parameter $\Delta$ and the orthorhombicity $\delta \equiv (a-b)/(a+b)$:

$$F[\Delta, \delta] = \frac{C}{2}\delta^2 + \frac{D}{4}\delta^4 - \frac{\alpha}{2}|\Delta|^2 + \frac{\beta}{4}|\Delta|^4 + \gamma|\Delta|^2\delta^2 \quad (6)$$

Here, we assume that the superconducting order parameter transforms under the tetragonal point symmetry, i.e., it does not break the $C_4$ rotational symmetry of the lattice. Since the lattice-nematic order parameter breaks this symmetry, the coupling to superconductivity is quadratic in $\delta$. Above, the coefficient $C$ is in fact the elastic shear modulus $C_{66}$, which is the inverse of the nematic susceptibility. The latter has a Curie–Weiss behavior (see Fig. 4 in the main text):

$$\chi_{nem} = \frac{1}{C_{66}} = \frac{1}{C_{66}^{(0)}}\frac{T^*}{T - T^*} \quad (7)$$

Here, $C_{66}^{(0)}$ is the "bare" value of shear modulus in the absence of nematic transition. Note that, the above formula can been derived rigorously from an effective model of lattice orthorhombicity $\delta$ coupled with an electronic nematic order parameter[29]. Here, we simply take $T^*$ to be the phenomenological Curie–Weiss temperature extracted from fitting the $d$-spacing in Fig. 4e. Note that, if $T^*$ is positive (for $x < 0.016$), we identify it with the nematic transition temperature $T_s$ such that $0 > C = -|C|$ is below $T_s$.

Minimizing this effective action with respect to the two-order parameters $\partial F/\partial\Delta = 0 = \partial F/\partial\delta$ we obtain in the mixed state with $T < \{T_s, T_c\}$ nonzero values of both parameters:

$$\Delta^2 = \frac{\alpha D - 2\gamma|C|}{\beta D - 4\gamma^2} = \frac{|\Delta_0|^2 - \left(\frac{2\gamma}{\beta}\right)\delta_0^2}{1 - \frac{4\gamma^2}{\beta D}} \quad (8)$$

$$\delta^2 = \frac{\beta|C| - 2\gamma\alpha}{\beta D - 4\gamma^2} = \frac{|\delta_0|^2 - \left(\frac{2\gamma}{D}\right)\Delta_0^2}{1 - \frac{4\gamma^2}{\beta D}}, \quad (9)$$

where, $\Delta_0 = \sqrt{\alpha/\beta}$ and $\delta_0 = \sqrt{|C|/D}$ are the values of the order parameters in the absence of coupling between them. In the coexistence phase, the free energy becomes:

$$F = F_{SC}^{(0)} - \frac{1}{4}\left(|C| - 2\gamma\frac{\alpha}{\beta}\right)\delta^2 = F_{SC}^{(0)} - \frac{D}{4}\delta^2\left(1 - \frac{4\gamma^2}{\beta D}\right), \quad (10)$$

where, $F_{SC}^{(0)} = -\alpha|\Delta_0|^2/4$. Note that, for the coexistence phase to be stable, the last term in the above expression must be positive, which is only possible if $\frac{4\gamma^2}{\beta D} < 1$, or equivalently, $\beta D > 4\gamma^2$.

There is no perceptible change in the superconducting transition temperature below $T_s$, implying $|\Delta| \simeq |\Delta_0|$. Substituting this into Eq. (8), we obtain:

$$\frac{2\gamma}{\beta}\delta_0^2 \ll |\Delta_0|^2 \quad (11)$$

By contrast, the suppression of the orthorhombicity below $T_c$ is substantial, $\delta \approx 0.5\delta_0$ (see Fig. 3b, c), meaning that $\left(\frac{2\gamma}{D}\right)|\Delta_0|^2 \approx \delta_0^2$ from Eq. (9). Substituting this into Eq. (11), we obtain:

$$\frac{4\gamma^2}{\beta D} \ll 1, \quad (12)$$

in other words, we can approximate the denominator in Eqs. (8) and (9) to be 1. This is also consistent with the requirement from Eq. (10) for the coexistence phase to be stable.

In summary, the phenomenological Landau free energy explains qualitatively the experimental data in the coexistence phase of superconductivity and nematicity. Furthermore, comparison with the experiment allows us to impose strong condition on the smallness of the coupling constant $\gamma$ in terms of inequality (12).

**Coexistence of three phases**. Below $x > 0.012$, NaFe$_{1-x}$Ni$_x$As has a long-range AF order, and the free energy in Eq. (10) has to be modified to include the magnetic order parameter $M$:

$$F_3[M, \Delta, \delta] = F[\Delta, \delta] - \frac{a}{2}M^2 + \frac{b}{4}M^4 - \mu|\delta| \cdot M^2 + w|\Delta|^2M^2, \quad (13)$$

where, we have included phenomenological coupling constants $\mu$ and $w$. The sign of $w$ is positive, in accord with our experimental observation that AF order and superconductivity compete with each other (see Fig. 2g, h in the main text). The sign in front of $\mu$ on the other hand is negative, indicating magnetoelastic coupling that favors the coexistence of magnetism and orthorhombic distortion. Because of this coupling, it is clear that $\delta$ will acquire an additional component proportional to $M^2$ inside the AF phase:

$$\delta = \delta(M = 0) + \kappa M^2 \qquad (14)$$

because $M^2$ and $|\Delta|^2$ repel each other via the last term in Eq. (13), this implies, in view of Eq. (14), that a new term proportional to $\Delta F \propto |\delta||\Delta|^2$ will be generated in the action, coupling the square of the superconducting order parameter linearly to the lattice orthorhombicity.

Working with full free energy in Eq. (13) is impractical because of the large number of phenomenological parameters that are difficult to determine experimentally. Nevertheless, it offers a qualitative insight into the coexistence between AF, lattice nematicity, and superconductivity, as the above discussion shows.

As a parenthetical remark, we note that the term $-\mu|\delta| \cdot M^2$ in free energy may appear surprising at first sight, as one might have expected that lattice distortion and magnetization should couple biquadratically. The reason for linear coupling is because the stripe AF order in iron pnictides breaks the lattice $C_4$ symmetry, as does the shear strain $\delta$[29,55–57]. Note that this conclusion holds independently of whether the microscopic origin of nematicity is purely magnetic[55,56] or due to orbital ordering of Fe $d_{xz}/d_{yz}$ orbitals[57–60].

**Data availability**. All data needed to evaluate the conclusions in the paper are present in the paper and/or the Supplementary Materials. Additional data related to this paper may be requested from the authors.

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

## Acknowledgments

The single crystal growth and neutron scattering work at Rice is supported by the U.S. DOE, BES under contract no. DE-SC0012311 (P.D.). A part of the material's synthesis and characterization work at Rice is supported by the Robert A. Welch Foundation Grant Nos. C-1839 (P.D.) and C-1818 (A.H.N.). A.H.N. also acknowledges the support of the US National Science Foundation Grant No. DMR-1350237. C.D.C. acknowledges the financial support by the NSFC (51471135), the National Key Research and Development Program of China (2016YFB1100101), Shenzhen Science and Technology Program (JCYJ20170815162201821), and Shaanxi International Cooperation Program (2017KW-ZD-07). We acknowledge the support of the High Flux Isotope Reactor, a DOE Office of Science User Facility operated by ORNL in providing the neutron research facilities used in this work.

## Author contributions

Single crystal growth and neutron scattering experiments were carried out by W.W., Y.S., C.C., and Y.L. with assistance from K.F.T., T.K., L.W.H., W.T., S.C., and P.D. Theoretical understandings were performed by R.Y. and A.H.N. The entire project is overseen by P.D and C.C. The paper was written by P.D., W.W., Y.S., and A.H.N. and all authors made comments.

## Additional information

**Competing interests:** The authors declare no competing interests.

