## [Peer Review File · Nature Communications]

Reviewers' comments:

Reviewer #1 (Remarks to the Author):

The authors present a study of $\text{Na}(\text{Fe}_{1-x}\text{Ni}_x)\text{As}$ reporting a detailed neutron scattering measurements of samples with $0 \leq x \leq 0.02$. They find that similar to other iron-based superconductors and in agreement with a previous report on $\text{NaFe}_{1-x}\text{Ni}_x\text{As}$ [25], $\text{NaFe}_{1-x}\text{Ni}_x\text{As}$ exhibits antiferromagnetic order setting in below a phase transition from a high-temperature tetragonal to a low-temperature orthorhombic structure. With Ni doping, both T_N and T_s are suppressed and a superconducting phase emerges for $x \geq 0.006$. The magnetic transition temperature is characterized from measurements of the magnetic superlattice peak (T_N), whereas the structural transition is deduced from high-resolution neutron diffraction and neutron Larmor diffraction experiments. In particular, the latter measurements reveal precursor effects on cooling to the structural phase transition which persist well into the doping range, where no full structural phase transition is observed anymore. The authors argue that their results are evidence of strong nematic fluctuations that may enhance superconductivity.

Overall, this is a very detailed neutron scattering investigation and highlights the unique capability of neutron scattering techniques to identify all relevant transition temperatures and the corresponding fluctuations in iron-based superconductors. It seems that neutron scattering experiments have been carried out carefully in well-characterized samples. The data analysis is well explained and can be easily followed. As far as I know this is indeed the first report on the determination of nematic fluctuations by scattering, in particular without applying any additional strain to the sample and in that an important step forward.

However, what I do not agree to is that the presence of local orthorhombic fluctuations is all that surprising as claimed by the authors:

On general terms, it would be naive to expect no (here orthorhombic) fluctuations above the transition temperature of a structural transition.

More specifically, the inset in Figure 4(f) of Ref. 25 shows a clear broadening of a powder diffraction line in an overdoped sample, $x = 0.025$ (see right panel of the figure attached), and the estimated upper limit for a persisting orthorhombic distortion of $b_0/a_0 \leq 3 \times 10^{-4}$ is well in line with the here presented results for $x \leq 0.02$. This diffraction line corresponds to a Bragg reflex, which splits on entering the orthorhombic phase in lower-doped samples. Another one, which does not split, shows also no broadening in the over-doped sample. Hence, there was clear evidence of orthorhombic distortions.

Yet, it is the achievement of the authors to have used scattering from single crystals to systematically determine the effects of nematic fluctuations in an iron-based superconductor. While Larmor diffraction using polarized neutron scattering is a rather exotic method, the clear results presented here can be a show case and motivate further studies with different techniques. Values in the range of $\Delta d/d \sim 10^{-4}$ can be determined also by, e.g., single-crystal diffraction, a much more common technique and able to probe samples of much smaller volume.

Hence, I would support a publication of the presented manuscript in Nature communications, in principle, if the authors give proper credit to previous results and discuss the impact of their results, e.g., with regard to simpler scattering techniques applicable to a wide range of materials.

Apart from the general remarks above I have a few comments and questions, which I would like the authors to consider in their revised manuscript:

Page2: Here and throughout the text, the authors do not mention at all the inelastic neutron scattering data presented in the supplementary material. Hence, it is apparently not important and can be removed. Otherwise, one should mention these results at the proper place in the manuscript.

Fig. 3: What are the lines – just guides to the eyes?

Figures 3e,f – Figures 4c,d: Curie-Weiss fits well for the data in Figure 4 but not for the same doping levels presented in Figure 3. Why is that? How were the measurements done? On different samples, experiments,...?

Page 6: I do not see a hump of T_c around the putative quantum-critical point. In fact, the T_c seems to be largely unaffected by anything “nematic” going on for $x \geq 0.008$. For instance, Δ/d decreases significantly from $x = 0.015$ to 0.02 (Figure 4) but T_c does not change significantly. Hence, the authors should drop their claim of nematic fluctuations enhancing superconductivity (see abstract) or substantiate it.

Reviewer #2 (Remarks to the Author):

I like this paper very much and think it is important. I have jotted down a lot of thoughts as I have read the paper which I include as suggestions for improvements to the authors. However, the basic point is that the results are (as far as I know) new, significant, and with the exceptions of some details, clearly and professionally presented. The results shed light on some of the central issues of the field. The paper should surely be accepted for publication.

Here are my more detailed comments and suggestions:

1) I am a little confused about a point – and as it is quite important I suggest that the authors make changes to the manuscript to make this clearer. It may be that this is covered in some subtle way, but the authors should keep in mind that their paper should ideally be written so that even quick reading reveals the important points. The issue here is that I think Δ is being used to signify two conceptually different quantities, although as the authors explain they are difficult in practice to distinguish under present circumstances – one is a resolved orthorhombic splitting of a peak and the other is a broadening of a peak that might be an unresolved splitting and might truly be simply a broadening due to local structure. This is particularly confusing when looking at the figures, where for example in Fig. 3, Δ is explicitly identified as an orthorhombic distortion (which a casual reader would interpret as meaning that an orthorhombic splitting has been resolved of the reported magnitude) but this is non-zero even in the absence of a structural transition. I understand that the suggested interpretation is that above T_s or where no clear T_s is observed, Δ should be interpreted as a broadening and below it should be interpreted as a splitting, but it is easy to get confused by this figure. Also, not included in the figure is any information about where the splitting is resolved and where it is not. I would suggest considering the following: Suppose two separate symbols are defined - Δ and $\tilde{\Delta}$. Δ can be used for where a clear splitting is observed and $\tilde{\Delta}$ for where it is just a broadening. I assume that at T 's well below T_s , there is a clear splitting seen – is this correct? IN this figure one could use open symbols for $\tilde{\Delta}$ and filled symbols for Δ .

2) A related question is what is the difference between Δ/d and Δ . I guess they are two different ways of fitting the data. In one, it is interpreted as two split peaks each of finite width, such that it gives an answer whether two resolved peaks can be seen in the data or not. Then Δ/d is some way of fitting it assuming a single peak. Maybe it is the full width at half maximum. Anyway, it would be very helpful to have a statement about precisely how these two quantities are defined from the same data.

3) A related question: Here T_s is identified as a kink in the T dependence of Δ , which is reasonable. However, there are important matters of interpretation that are implicit here, so anything that can corroborate this is significant. For instance, if there is a transition at T_s there should be thermodynamic signatures – for instance specific heat or thermal expansivity or the like. Surely there are elastic measurements that can identify a true orthorhombic transition. Since there are likely to be domains below T_s , there should also be some degree of history dependence of experimental quantities cycled through T_s in the presence of strain or a symmetry breaking (in plane) magnetic field. Are there elasto-resistance measurements on this material? As they are (as far as I know) the most sensitive measurement of nematic symmetry breaking is in the elasto-resistance, so correlating T_s as determined that way with the present results would be useful.

- 4) "These observations are either reflective of an intrinsic rotational-symmetry-broken phase [12–15] or simply result from a large nematic susceptibility [16–18]." This is an important point. Firstly, I think that the possibility that the results simply reflect a large susceptibility was first made by Fisher and collaborators. More importantly, there is another logical possibility that was pointed out (in a talk at least – not sure whether it is in print) by Orenstein: There could be a surface ("extraordinary") transition in which the C4 symmetry is broken at the surface at a higher temperature than in the bulk. Such things can happen in CDW systems. I am not sure what the status of this proposal is, but it is an interesting proposal that should not be discarded out of hand.
- 5) The evolution of the magnetism in the 0.012 sample is just plain cool. I wanted to say that.
- 6) Concerning the Currie Weiss behavior of the nematicity above T_s : It might be worth noting that there are now at least two distinct theoretical lines of analysis that predict such a behavior. The first is the DQMC results reported in Schattner et al, PRX 6, 31028 (2016), on the neighborhood of a nematic QCP where it is shown that the nematic susceptibility (for reasons that I think are still unclear) very accurately obeys a Currie-Weiss law. There is a second line of reasoning, by Schmalian and collaborators, that invokes the effect of long-range strain coupled to the nematic order parameter to produce mean-field like behavior, and thus to account for the Currie Weiss behavior previously reported from elasto-resistance.
- 7) One important point that is clear in the authors data that is not emphasized is that there is a distinct magnetic and nematic QCP. There have been both theoretical and experimental arguments adduced that argue for a convergence of the two transition leading to a single, first order, zero temperature critical point at which both nematic and magnetic order onset. It seems to me pretty clear from the present data that the nematic QCP occurs at an unambiguously higher critical doping concentration, and this should probably be highlighted.
- 8) Another point I am very confused about is time scales. The local nematic order that is being reported here – should I think of this as static patterns of distortion or can I think of them as slowly fluctuating in time? Do we have any estimate of the time-scales over which they are effectively static or fluctuating? Among other things, this affects what I might think is the role of quenched disorder in producing the observed spectra. This may be my most significant complaint about the paper – that this issue was not addressed at all.

Reviewer #3 (Remarks to the Author):

This paper reports neutron diffraction data, including neutron Larmor diffraction, on $\text{NaFe}_{1-x}\text{Ni}_x\text{As}$ with several Ni concentrations covering a wide range of x-T phase space. They show that local orthorhombic lattice distortions exist in the underdoped and overdoped regions. Their data clearly show that in the underdoped region the local distortions persist well above the tetragonal-to-orthorhombic structural transition and magnetic order. Even in the overdoped region with no magnetic and no structural transitions, the local distortions persist. By fitting their T-dependence to the Curie-Weiss law, they drew the connection to a local static nematic order.

Their data and analysis are good, and their conclusion is reasonable based on the data and analysis. I recommend the publication of this work in Nature Comm.

Our detailed point-by-point replies to referees' comments:

Referee #1:

"The authors present a study of $\text{Na}(\text{Fe}_{1-x}\text{Ni}_x)\text{As}$ reporting a detailed neutron scattering measurements of samples with $0 \leq x \leq 0.02$. They find that similar to other iron-based superconductors and in agreement with a previous report on $\text{NaFe}_{1-x}\text{Ni}_x\text{As}$ [25], $\text{NaFe}_{1-x}\text{Ni}_x\text{As}$ exhibits antiferromagnetic order setting in below a phase transition from a high-temperature tetragonal to a low-temperature orthorhombic structure. With Ni doping, both T_N and T_s are suppressed and a superconducting phase emerges for $x \geq 0.006$. The magnetic transition temperature is characterized from measurements of the magnetic superlattice peak (T_N), whereas the structural transition is deduced from high-resolution neutron diffraction and neutron Larmor diffraction experiments. In particular, the latter measurements reveal precursor effects on cooling to the structural phase transition which persist well into the doping range, where no full structural phase transition is observed anymore. The authors argue that their results are evidence of strong nematic fluctuations that may enhance superconductivity. "

These are accurate description of our results.

"Overall, this is a very detailed neutron scattering investigation and highlights the unique capability of neutron scattering techniques to identify all relevant transition temperatures and the corresponding fluctuations in iron-based superconductors. It seems that neutron scattering experiments have been carried out carefully in well-characterized samples. The data analysis is well explained and can be easily followed. As far as I know this is indeed the first report on the determination of nematic fluctuations by scattering, in particular without applying any additional strain to the sample and in that an important step forward."

Again, these are accurate description of the paper.

"However, what I do not agree to is that the presence of local orthorhombic fluctuations is all that surprising as claimed by the authors:

On general terms, it would be naïve to expect no (here orthorhombic) fluctuations above the transition temperature of a structural transition.

More specifically, the inset in Figure 4(f) of Ref. 25 shows a clear broadening of a powder diffraction line in an overdoped sample, $x = 0.025$ (see right panel of the figure attached), and the estimated upper limit for a persisting orthorhombic distortion of $b_0/a_0 \leq 3 \times 10^{-4}$ is well in line with the here presented results for $x \leq 0.02$. This diffraction line corresponds to a Bragg reflex, which splits on entering the orthorhombic phase in lower-doped samples. Another one, which does not split, shows also no broadening in the over-doped sample. Hence, there was clear evidence of orthorhombic distortions."

We appreciate very much these comments, and agree completely with the description of the referee on previous work. In the revised draft, we have made it very clear that previous work have already shown signatures of orthorhombic distortion in the overdoped sample. However, no detailed temperature and doping dependence of these measurements were carried out prior to this work. Clear credit is given to the authors of Ref. [25] (Ref. [26] in the current revised version) in the revised draft introduction.

“Yet, it is the achievement of the authors to have used scattering from single crystals to systematically determine the effects of nematic fluctuations in an iron-based superconductor. While Larmor diffraction using polarized neutron scattering is a rather exotic method, the clear results presented here can be a show case and motivate further studies with different techniques. Values in the range of $\Delta d/d \sim 10^{-4}$ can be determined also by, e.g., single-crystal diffraction, a much more common technique and able to probe samples of much smaller volume. ”

We agree with these statements.

“Hence, I would support a publication of the presented manuscript in Nature communications, in principle, if the authors give proper credit to previous results and discuss the impact of their results, e.g., with regard to simpler scattering techniques applicable to a wide range of materials.”

We appreciate these comments very much. In the revised draft, we give credit to previous work, and in the discussion section of the paper, also propose possible future diffraction experiment (albeit rather difficult) to see orthorhombic distortion in other materials.

“Page2: Here and throughout the text, the authors do not mention at all the inelastic neutron scattering data presented in the supplementary material. Hence, it is apparently not important and can be removed. Otherwise, one should mention these results at the proper place in the manuscript.”

These data are useful characterization of the $x=0.12$ sample, and is now mentioned in the revised draft of the paper.

“Fig. 3: What are the lines – just guides to the eyes?”

Yes. This has now been made clear in the revised draft.

“Figures 3e,f – Figures 4c,d: Curie-Weiss fits well for the data in Figure 4 but not for the same doping levels presented in Figure 3. Why is that? How were the measurements done? On different samples, experiments,...?”

We apologize for confusing the referee, the raw data from Larmor diffraction in Fig. 3e and 3f and Figs. 4c and 4d are the same, but we analyzed them slightly differently. We now explicitly state this in the caption of these figures. The data in Fig. 3e and 3f are analyzed by assuming an orthorhombic distortion, and then proceeding to extract Δ . This is done by taking Δ to be zero at a temperature well above any phase transitions (50 K). Details of the analysis are spelt out clearly in the caption of Fig. 3 as well as in Supplementary Note 3, which we clearly reference when discussing these results. Given Δ

extracted in Fig. 3 for these samples do not resemble an order parameter, we are motivated instead to analyze the results as a broadening ($\Delta d/d$) due to local orthorhombic distortion in Fig. 4c, d. The analysis in Fig. 4 using $\Delta d/d$ does not need a reference temperature and therefore allows us to use all available data to more reliably perform Curie-Weiss fits over a larger temperature range. The details and differences between the two methods are now clearly explained in Supplementary Note 3 and are referenced when these results are presented in the main text.

“: I do not see a hump of T_c around the putative quantum-critical point. In fact, the T_c seems to be largely unaffected by anything “nematic” going on for $x \geq 0.008$. For instance, $\Delta d/d$ decreases significantly from $x = 0.015$ to 0.02 (Figure 4) but T_c does not change significantly. Hence, the authors should drop their claim of nematic fluctuations enhancing superconductivity (see abstract) or substantiate it.”

We appreciate these comments. We have removed the mention of nematic fluctuations enhancing T_c from the abstract. In addition, we note that when we mention a ‘hump’ towards the end of the manuscript, there we are referring to the hump of magnetically ordered region in the T-x phase diagram, seen in Fig. 1(d), with a clear peak at T_c (rather than some value of x) due to competition between magnetic order and superconductivity. We have now improved the writing to convey the message more clearly.

Referee #2:

“I like this paper very much and think it is important. I have jotted down a lot of thoughts as I have read the paper which I include as suggestions for improvements to the authors. However, the basic point is that the results are (as far as I know) new, significant, and with the exceptions of some details, clearly and professionally presented. The results shed light on some of the central issues of the field. The paper should surely be accepted for publication.”

We appreciate these supportive comments very much.

“1) I am a little confused about a point – and as it is quite important I suggest that the authors make changes to the manuscript to make this clearer. It may be that this is covered in some subtle way, but the authors should keep in mind that their paper should ideally be written so that even quick reading reveals the important points. The issue here is that I think $\Delta d/d$ is being used to signify two conceptually different quantities, although as the authors explain they are difficult in practice to distinguish under present circumstances – one is a resolved orthorhombic splitting of a peak and the other is a broadening of a peak that might be an unresolved splitting and might truly be simply a broadening due to local structure. This is particularly confusing when looking at the figures, where for example in Fig. 3, $\Delta d/d$ is explicitly identified as an orthorhombic distortion (which a casual reader would interpret as meaning that an orthorhombic splitting has been resolved of the reported magnitude) but this is non-zero even in the absence of a structural transition. I understand that the suggested interpretation is that above T_s or where no clear T_s is observed, $\Delta d/d$ should be interpreted as a broadening and below it should be interpreted as a splitting, but it is easy to get confused by this figure. Also, not included in the figure is any information about where the splitting is resolved and where it is not. I would suggest considering the following: Suppose two separate symbols are defined - $\Delta d/d$ and $\tilde{\Delta d/d}$. $\Delta d/d$ can be used for where a clear splitting is observed and $\tilde{\Delta d/d}$ for where it is just a broadening. I assume that at T_s

well below T_s , there is a clear splitting seen – is this correct? IN this figure one could use open symbols for $\tilde{\Delta}$ and filled symbols for Δ .”

We appreciate very much these suggestions to improve the figure and clarify the notation. In the revised figures, we have used different symbols for points where a splitting can be directly observed compared to those with just a broadening, as suggested by the referee. We also discuss the technical aspects of the fitting and limitations in Supplementary Note 3, and referenced it in the main text where appropriate.

“2) A related question is what is the difference between Δ and $\tilde{\Delta}$. I guess they are two different ways of fitting the data. In one, it is interpreted as two split peaks each of finite width, such that it gives an answer whether two resolved peaks can be seen in the data or not. Then Δ is some way of fitting it assuming a single peak. Maybe it is the full width at half maximum. Anyway, it would be very helpful to have a statement about precisely how these two quantities are defined from the same data.”

Yes. The referee is correct. In figure 3, we assumed two peaks and in Fig. 4, a broadening is used. This is essentially the same question as raised by referee 1, which we have now clarified in the revised draft. We have also improved the schematic in Fig. 1(c) to directly convey the meaning of the two quantities. Specifically, we added a symbol for $\tilde{\Delta}$ in Fig. 1(c) to clarify its meaning.

“3) A related question: Here T_s is identified as a kink in the T dependence of Δ , which is reasonable. However, there are important matters of interpretation that are implicit here, so anything that can corroborate this is significant. For instance, if there is a transition at T_s there should be thermodynamic signatures – for instance specific heat or thermal expansivity or the like. Surely there are elastic measurements that can identify a true orthorhombic transition. Since there are likely to be domains below T_s , there should also be some degree of history dependence of experimental quantities cycled through T_s in the presence of strain or a symmetry breaking (in plane) magnetic field. Are there elasto-resistance measurements on this material? As they are (as far as I know) the most sensitive measurement of nematic symmetry breaking in the elasto-resistance, so correlating T_s as determined that way with the present results would be useful.”

The referee is correct that T_s is identified as a kink in T -dependence of Δ . Additional evidence for T_s comes independently from transport data shown in Supplementary Fig. S3 for samples prepared the same way in our lab. In the revised draft, we made this absolutely clear in the main text, citing Supplementary Fig. S3 in the main text to make this point clear. While elasto-resistance measurements would be desirable, we have not carried these out and are not aware of any elasto-resistance measurements on this system.

“4) “These observations are either reflective of an intrinsic rotational-symmetry-broken phase [12–15] or simply result from a large nematic susceptibility [16–18].” This is an important point. Firstly, I think that the possibility that the results simply reflect a large susceptibility was first made by Fisher and collaborators. More importantly, there is another logical possibility that was pointed out (in a talk at least – not sure whether it is in print) by Orenstein: There could be a surface (“extraordinary”) transition in which the C_4 symmetry is broken at the surface at a higher temperature than in the bulk. Such things can happen in CDW systems. I am not sure what the status of this proposal is, but it is an interesting proposal that should not be discarded out of hand.”

We thank the referee for the note. We agree that insofar as the enhancement of the nematic susceptibility is concerned, credit should be given to Fisher *et al*, and therefore have added their paper to the present sentence (Ref. 20 in the current version). In terms of the possibility raised by Orenstein, we know of his group's work on arXiv (Ref. 16), but were unable to find a publication of the results. In the revised draft, this scenario is explicitly mentioned in the introduction. However, we note that our neutron scattering work probes the bulk, so a surface effect would not be detected, and hence does not apply to our case.

"5) The evolution of the magnetism in the 0.012 sample is just plain cool. I wanted to say that."

We appreciate this very much, and this is really the interesting aspect of these materials.

*"6) Concerning the Currie Weiss behavior of the nematicity above T_s : It might be worth noting that there are now at least two distinct theoretical lines of analysis that predict such a behavior. The first is the DQMC results reported in Schattner *et al*, PRX 6, 31028 (2016), on the neighborhood of a nematic QCP where it is shown that the nematic susceptibility (for reasons that I think are still unclear) very accurately obeys a Currie-Weiss law. There is a second line of reasoning, by Schmalian and collaborators, that invokes the effect of long-range strain coupled to the nematic order parameter to produce mean-field like behavior, and thus to account for the Currie Weiss behavior previously reported from elasto-resistance."*

In the revised draft, we discuss and cite the above mentioned works.

"7) One important point that is clear in the authors data that is not emphasized is that there is a distinct magnetic and nematic QCP. There have been both theoretical and experimental arguments adduced that argue for a convergence of the two transition leading to a single, first order, zero temperature critical point at which both nematic and magnetic order onset. It seems to me pretty clear from the present data that the nematic QCP occurs at an unambiguously higher critical doping concentration, and this should probably be highlighted."

Yes. A unique advantage of this system is that magnetic order has a well-defined onset temperature, making it ideal to address this question. The well-split T_S and T_N we observe up to $x=0.012$ implies separate nematic and magnetic QCPs, in the absence of superconductivity. We have emphasized this point in the revised draft of the paper, both mentioning it in the abstract as well as in the main text.

"8) Another point I am very confused about is time scales. The local nematic order that is being reported here – should I think of this as static patterns of distortion or can I think of them as slowly fluctuating in time? Do we have any estimate of the time-scales over which they are effectively static or fluctuating? Among other things, this affects what I might think is the role of quenched disorder in producing the observed spectra. This may be my most significant complaint about the paper – that this issue was not addressed at all."

Neutron scattering, like other scattering techniques, has a non-zero energy resolution. In the present experiment, we would expect that the energy resolution to be about 1 meV since Heusler crystal used to make monochromatic beam has d-spacing similar to pyrolytic graphite and thus similar energy resolution. This means that the scattering we measure includes true elastic scattering as well as quasi-static fluctuations slower than a time scale of $\hbar/2\Delta E \sim 0.3$ ps. This issue is now addressed in the paper, as well as some references concerning the technique (Physica B 406, 2333 (2011), and PRL 85,

2553 (2000). We agree with the referee that quenched disorder that produces internal strain very likely plays a role in what we observe, e.g. possibly acting to pin the distortions in the absence of an external strain. In the revised manuscript, we have made this point more clear. If the distortions are due to pinning by quenched disorder, then very likely the distortions are static. It should be noted that what is seen in the pure elastic channel, is the average structure over time. While an orthorhombic distortion may fluctuate to the domain rotated by 90 degrees from time to time, quenched disorder will favor one over the other, and when averaged over time will give rise to orthorhombic distortions seen in the static channel. Therefore our results should be interpreted as being elastic or quasi-elastic (not a delta function in energy, but rather a peak with non-zero width), rather than purely fluctuations.

Referee #3:

We appreciate very much the referee's comments and recommendation for publication.

REVIEWERS' COMMENTS:

Reviewer #1 (Remarks to the Author):

The authors have addressed all criticisms raised by the referees and I support publication of the manuscript in its current form.

Reviewer #2 (Remarks to the Author):

The paper was acceptable for publication previously and has been significantly improved. I recommend accepting it.

Since I have read the paper through carefully, I cannot refrain from making a number of very minor observations for the OPTIONAL consideration of the authors.

In the second paragraph - the discussion of the proposal that there is a true bulk nematic phase even where no structural evidence has been seen leaves out one important point. The issue is in this case, how does one explain the existence of a sharp phase transition from the weakly nematic phase at high T and/or large x to a detectably orthorhombic phase below T_s . In general, I think this is not plausible. The only way that this would be possible as far as I know is if in addition to the nematic symmetry breaking, there is some other aspect of the symmetry that distinguishes the two putative nematic phases. My guess is that the evidence of this mysterious nematic phase is simply WRONG. This is not the proposal of the present authors, so this is not an important point for the present paper. However, if they could make any stronger or clearer statements about this, it would be useful. At any rate, they might want to point out the problem that the existence of a sharp phase transition at T_s presents for this scenario.

I do not understand the statement following Eq (2) that the SC order parameter itself remains unchanged. I would think that the biquadratic coupling γ would imply similar effects on the SC order parameter, albeit proportional to $(\gamma/\beta)|\delta|^2$.

The statements that there is a "QCP near $x \geq 0.015$ " and "changes from positive to negative near $x \geq 0.015$ " do not make sense. $x=1$ is also ≥ 0.015 . I think what is meant is that there is a "QCP for $x=x_c$ where $x_c \approx 0.015$, or possibly slightly larger." or if you want " $x_c \gtrsim 0.015$ ".

The discussion starting with "Alternatively, the observed Currie Weiss temperature dependence ..." is a bit confusing. The difference in the proposals in [43] and [44] is not particularly whether the nematic is spin-driven or not. Both arguments concern effective field theories that really are not all the sensitive to the microscopics. The distinction is that [43] does not include the coupling of the nematic order to long-range strain fields, while [44] emphasizes the role of long-range strain in stabilizing mean-field exponents. I am not suggesting changing the structure of this discussion at all - just making clearer what the distinction is.

Reviewer #1 (Remarks to the Author):

“The authors have addressed all criticisms raised by the referees and I support publication of the manuscript in its current form.”

We thank the referee’s recommendation to publish our work.

Reviewer #2 (Remarks to the Author):

“The paper was acceptable for publication previously and has been significantly improved. I recommend accepting it.

Since I have read the paper through carefully, I cannot refrain from making a number of very minor observations for the OPTIONAL consideration of the authors.”

We thank the referee’s recommendation to publish our work, and address the referee’s optional revisions below.

“In the second paragraph - the discussion of the proposal that there is a true bulk nematic phase even where no structural evidence has been seen leaves out one important point. The issue is in this case, how does one explain the existence of a sharp phase transition from the weakly nematic phase at high T and/or large x to a detectably orthorhombic phase below T_s . In general, I think this is not plausible. The only way that this would be possible as far as I know is if in addition to the nematic symmetry breaking, there is some other aspect of the symmetry that distinguishes the two putative nematic phases. My guess is that the evidence of this mysterious nematic phase is simply WRONG. This is not the proposal of the present authors, so this is not an important point for the present paper. However, if they could make any stronger or clearer statements about this, it would be useful. At any rate, they might want to point out the problem that the existence of a sharp phasetransition at T_s presents for this scenario.”

We thank the referee for pointing this out, in the revised manuscript we have noted in the second paragraph that in this scenario there is no additional symmetry-breaking below T_s despite the sharp increase of the nematic order parameter.

“I do not understand the statement following Eq (2) that the SC order parameter itself remains unchanged. I would think that the biquadratic coupling γ would imply similar effects on the SC order parameter, albeit proportional to $(\gamma/\beta) |\delta|^2$.”

Yes. The referee is correct that changes in Δ is related to δ^2 as shown in eq. 8 in method section (now). Our point was because changes in δ is so small, Δ in realistic sense does not change much at all. We added a statement to say this explicitly in the revised draft.

"The statements that there is a "QCP near $x \geq 0.015$ " and "changes from positive to negative near $x \geq 0.015$ " do not make sense. $x=1$ is also ≥ 0.015 . I think what is meant is that there is a "QCP for $x=x_c$ where $x_c \approx 0.015$, or possibly slightly larger." or if you want " $x_c \sim 0.015$ "."

We thank the referee for pointing this out, and have revised the manuscript following the referee's suggestion.

"The discussion starting with "Alternatively, the observed Currie Weiss temperature dependence ..." is a bit confusing. The difference in the proposals in [43] and [44] is not particularly whether the nematic is spin-driven or not. Both arguments concern effective field theories that really are not all that sensitive to the microscopics. The distinction is that [43] does not include the coupling of the nematic order to long-range strain fields, while [44] emphasizes the role of long-range strain in stabilizing mean-field exponents. I am not suggesting changing the structure of this discussion at all - just making clearer what the distinction is."

We appreciate these statements from the referee and did not change the paper as suggested by the referee.